# Effect of Arginine on Chaperone-Like Activity of HspB6 and Monomeric 14-3-3ζ

**DOI:** 10.3390/ijms21062039

**Published:** 2020-03-16

**Authors:** Valeriya V. Mikhaylova, Tatiana B. Eronina, Natalia A. Chebotareva, Vladimir V. Shubin, Daria I. Kalacheva, Boris I. Kurganov

**Affiliations:** Bach Institute of Biochemistry, Federal Research Centre “Fundamentals of Biotechnology” of the Russian Academy of Sciences, Leninsky pr. 33, 119071 Moscow, Russia

**Keywords:** HspB6, monomeric 14-3-3ζ, anti-aggregation activity, arginine

## Abstract

The effect of protein chaperones HspB6 and the monomeric form of the protein 14-3-3ζ (14-3-3ζ_m_) on a test system based on thermal aggregation of UV-irradiated glycogen phosphorylase *b* (UV-Ph*b*) at 37 °C and a constant ionic strength (0.15 M) was studied using dynamic light scattering. A significant increase in the anti-aggregation activity of HspB6 and 14-3-3ζ_m_ was demonstrated in the presence of 0.1 M arginine (Arg). To compare the effects of these chaperones on UV-Ph*b* aggregation, the values of initial stoichiometry of the chaperone–target protein complex (*S*_0_) were used. The analysis of the *S*_0_ values shows that in the presence of Arg fewer chaperone subunits are needed to completely prevent aggregation of the UV-Ph*b* subunit. The changes in the structures of HspB6 and 14-3-3ζ_m_ induced by binding of Arg were evaluated by the fluorescence spectroscopy and differential scanning calorimetry. It was suggested that Arg caused conformational changes in chaperone molecules, which led to a decrease in the thermal stability of protein chaperones and their destabilization.

## 1. Introduction

Protein aggregation is a universal and unfavorable process for all cells, leading to production of non-native protein structures. Accumulation of toxic aggregates in cells is associated with a number of neurodegenerative diseases [1,2,3,4,5,6]. However, cells have protective mechanisms that allow them to minimize the risk of protein aggregation [7,8]. Small heat shock proteins (sHsps) belong to the family of molecular chaperones and protect cells against different kind of stresses, serving as a first line of defense in preventing protein aggregation [9,10,11]. One of the most important functions of sHsps is to prevent protein aggregation by binding of non-native or misfolded protein molecules and hold them in a folding-competent conformation. In addition, sHsps are involved in the regulation of many cellular processes and assist in the maintenance of protein homeostasis [10].

One of the representatives of sHsps family is a small HspB6 protein with a molecular mass of subunit 17.2 kDa. Just like the other members of this family, the HspB6 monomer contains conservative α-crystallin-, flexible N- and variable C-domains. At neutral pH and physiological ionic strength recombinant human HspB6 formed small oligomers with an apparent molecular weight of 55–60 kDa, representing dimers and tetramers [12]. This is not typical for sHsps, because as a rule, these proteins tend to form large polydisperse assemblies ranging from 10-mer to 40-mer and higher, having very dynamic structures [13,14,15]. All oligomeric forms possess chaperone-like activity and easily exchange their subunits. It is believed that a flexible dynamic quaternary structure is necessary for sHsps activity [10]. In addition to suppressing protein aggregation under stress conditions, HspB6 performs many other features. HspB6 is able to interact with many cellular proteins, thus participating in the processes of muscle relaxation, apoptosis, cell migration and reticular stress [16,17,18,19,20].

Among other important proteins that regulate various processes in the cell one can point to the family of regulatory proteins 14-3-3, which are universally distributed in eukaryotic cells. The 14-3-3 proteins interact with hundreds of proteins predominantly phosphorylated and also participate in regulation of the important cellular processes such as apoptosis, cell division, transcription and in functioning cell signaling pathways [21,22]. 14-3-3 proteins primarily exist as homo- or hetero-dimers (for example, it forms heterodimers with HspB6), which can dissociate under certain conditions [22]. Under unfavorable cellular conditions the equilibrium between the dimeric and monomeric forms can be shifted towards monomeric forms [22]. There are 7 isoforms of human 14-3-3 proteins [23], some of isoforms (ε and γ) are dimeric, whereas other isoforms exist as a mixture of dimer/monomer [23]. It was reported that phosphorylation of Ser58 of certain 14-3-3 isoforms (η, β, and ζ) in response to activation of different pathways could induce partial dissociation of 14-3-3 dimer (see a review [22]). The structure and functions of the 14-3-3 dimer are well characterized, while much less is known about the properties of the monomeric form of 14-3-3. Earlier it was shown that monomeric 14-3-3ζ has a chaperone-like activity and is stabilized by phosphorylated HspB6 [24]. Previously, using different target proteins, we found that monomeric form of 14-3-3ζ might have comparable or even higher chaperone-like activity than either the dimeric form of 14-3-3ζ or HspB6 and HspB5 [25].

Therefore, it was interesting to use a monomeric mutant of 14-3-3ζ (14-3-3ζ_m_) and HspB6 as protein chaperones for this work.

A chemical chaperone, arginine (Arg) is widely used in the biotechnology for suppressing processes of protein aggregation [26,27,28,29]. The role of Arg in specific and non-specific protein–protein interactions was reviewed in the works [30,31,32,33]. However, it should be noted that Arg can not only suppress, but also enhance the aggregation of the target protein [34,35,36]. In addition, Arg has been shown to regulate the anti-aggregation activity of some sHsps (α-crystallin, HspB5 and HspB4) [37,38,39]. It was suggested that this action of Arg may provide a protective mechanism against toxic aggregates in a living cell [39]. The effect of Arg on the chaperone activity of sHsp is highly dependent on the selection of target protein undergoing aggregation in test system [39]. In this regard, when studying the effect of Arg on chaperone-like activity of protein chaperones, it is also necessary to study the effect of Arg on target protein aggregation.

According to our current knowledge, there is not enough data on the effect of Arg on chaperone-like activity of HspB6 and the monomeric form of 14-3-3ζ. In order to study the effect of Arg on the anti-aggregation activity of HspB6 and engineered monomeric 14-3-3ζ_m_, we used a test system based on aggregation of UV-irradiated glycogen phosphorylase *b* (UV-Ph*b*) under physiological conditions (at 37 °C and an ionic strength of 0.15 M), which was described earlier [40]. The aggregation process for UV-Ph*b*, obeying the mechanism of nucleation-dependent aggregation, involves the rate-limiting stage of structural reorganization of UV-Ph*b* molecule, the nucleation stage and the stage of aggregate growth. The effect of the chemical chaperone Arg on this test system was also studied in detail earlier [36].

In the present work, we compared the anti-aggregation activities of protein chaperones HspB6 and engineered monomeric 14-3-3ζ_m_ in the absence and in the presence of 0.1 M Arg with UV-irradiated Ph*b* as a target protein and the stoichiometry of chaperone–UV-Ph*b* complexes were determined. The changes in the structures of HspB6 and monomeric form of 14-3-3ζ induced by binding of Arg were assessed by the fluorescence spectroscopy and differential scanning calorimetry.

## 2. Results

### 2.1. The Effect of Arg on the Chaperone-Like Activity of HspB6 and Monomeric 14-3-3ζ

For testing the anti-aggregation activity of protein chaperones HspB6 and monomeric 14-3-3ζ (14-3-3ζ_m_) in the absence and in the presence of 0.1 M Arg UV-irradiated Ph*b* (UV-Ph*b*) was used as a target protein. The studies were carried out using the method of dynamic light scattering (DLS) under physiological conditions: at a temperature of 37 °C and an ionic strength of 0.15 M. The aggregation process was initiated by the addition of UV-Ph*b* to the sample containing Arg, protein chaperones or its mixture. The initial part of the sigmoidal kinetic curve of UV-Ph*b* aggregation, representing the time dependence of the light scattering intensity *I*(*t*) (Figure 1A,B and Figure 3A,B), corresponds to the nucleation stage, i.e., the stage of assembling unfolded protein molecules into nuclei capable of further growth. To characterize the duration of the nucleation stage (*t**) and the initial rate of the stage of aggregate growth (*v_0_*), a polynomial of the second degree can be used to describe the part of the kinetic curve of aggregation above the inflexion point (see Appendix A):
(1)$$I=v_{0}\left(t-t^{*}\right)-B{\left(t-t^{*}\right)}^{2}$$
where *I* is the light scattering intensity, *t* is the time, *t** is a length on the abscissa axis cut off by the theoretical curve calculated with this equation and *B* is a constant. Parameter *t** can be considered as a characteristic of the nucleation stage rate. The lower the *t** value, the higher is the rate of the nucleation stage. 

The process of UV-Ph*b* aggregation proceeds through the formation of two types of aggregates with different hydrodynamic radius *R*_h_: large aggregates (*R*_h,2_) and smaller aggregates (*R*_h,1_) [40]. Larger aggregates with a hydrodynamic radius *R*_h_ = *R*_h,2_ increase in size faster and, accordingly, make the more significant contribution to the light scattering intensity of the protein solution. Figure 1 shows the dependences of the light scattering intensity (*I* − *I*_0_) and hydrodynamic radius (*R*_h,2_) on time for aggregation of UV-Ph*b* (0.25 mg/mL) in the presence of different concentrations of HspB6 (Figure 1A,C) and HspB6 + 0.1 M Arg (Figure 1B,D). The suppression of the light scattering intensity in time with increasing HspB6 concentration was observed in both cases and can be characterized by parameters *v*_0_ and *t**. In addition, an increase in the concentration of chaperone leads to the fact that the sizes of the hydrodynamic radii of the formed aggregates (*R*_h,2_) begin to increase later both in the absence and in the presence of Arg.

We can calculate the *R*_h,2_ values at *t* = *t**. These *R*_h,2_ values are designated as *R*_h_*. The value of *R*_h_* characterizes the size of nuclei formed by the time of the nucleation stage completion (*t**). The values of *R*_h_* for UV-Ph*b* in the absence of any additives and in the presence of 0.1 M Arg, HspB6 or Arg + HspB6 are presented in Table 1. It should be noted that the value of *R*_h_* for UV-Ph*b* in the presence of HspB6 was independent of chaperone concentration. The obtained data indicate that, the addition of HpsB6 to UV-Ph*b* leads to an increase in the size of the nuclei formed at *t* = *t**, but at the combined action of the chaperone and Arg the size of the nucleus decreases.

The anti-aggregation activity of protein chaperones can be characterized by the value of the initial stoichiometry of the chaperone–target protein complex (*S*_0_) and can be determined from the initial linear part of the dependence of *v*_0_ on the [chaperone]/[UV-Ph*b*] ratio [41]:(2)$$v_{0}=v_{0}{}^{\left(0\right)}\left(1-x/S_{0}\right)$$

In this equation *v*_0_ is the initial rate of aggregation for the stage of aggregate growth found from Equation (1), *x* is the ratio of molar concentration of chaperone calculated on subunit to molar concentration of UV-Ph*b* calculated on monomer with molecular mass of 97.4 kDa (*x* = [chaperone]/[UV-Ph*b*]) and *v*_0_^(0)^ is the *v*_0_ value at *x* = 0. The *S*_0_ value shows how many molecules of chaperone can be bound by a monomer of UV-Ph*b* in order to completely suppress UV-Ph*b* aggregation. The dimension of parameter *S*_0_ is (moles of chaperone monomer/moles of UV-Ph*b* monomer).

The parameter *S*_0_ was used to compare the effects of HspB6 on UV-Ph*b* aggregation in the absence and in the presence of 0.1 M Arg. To estimate the *S*_0_ values, the dependences of *v*_0_/*v*_0_^(0)^ on the HspB6/UV-Ph*b* ratio were constructed (Figure 2A). The fact that the relative value *v*_0_/*v*_0_^(0)^ is plotted on the ordinate axis means that the obtained dependence demonstrates the Arg effect precisely as a result of its binding directly to the chaperone. It was taken into account that the molecular weight of HspB6 subunit is equal to 17.2 kDa.

Using Equation (2) the initial stoichiometry of the HspB6–UV-Ph*b* complexes (*S*_0_) in the absence (curve 1) and in the presence of 0.1 M Arg (curve 2) was determined: *S*_0_ = 11.5 ± 0.8 (*R*^2^ = 0.910) and *S*_0_ = 8.2 ± 0.6 (*R*^2^ = 0.894), respectively. The received data mean that in the presence of Arg fewer HspB6 molecules are needed to completely suppress aggregation of UV-Ph*b*.

As can be seen from Figure 2B, the relative duration of the nucleation stage, *t**/*t**_0_, practically does not change with increasing HspB6/UV-Ph*b* ratio in the absence of Arg (curve 1) and slightly increases in the presence of 0.1 M Arg (curve 2). So, in the presence of Arg, the protective effect of HspB6 on UV-Ph*b* aggregation is enhanced.

Figure 3 shows the dependences of (*I* − *I*_0_) and *R*_h,2_ on time for aggregation of UV-Ph*b* (0.25 mg/mL) in the presence of different concentrations of 14-3-3ζ_m_ (Figure 3A,C) and 14-3-3ζ_m_ + 0.1 M Arg (Figure 3B,D).

The values of *R*_h_* obtained for UV-Ph*b* aggregation in the absence of any additives and in the presence of 0.1 M Arg, 14-3-3ζ_m_ or Arg + 14-3-3ζ_m_ are presented in Table 1. As in the case of HspB6, the value of *R*_h_* for UV-Ph*b* in the presence of 14-3-3ζ_m_ was independent on chaperone concentration. Based on the data obtained, we can conclude that 14-3-3ζ_m_ provokes an increase in the size of the nuclei formed at *t* = *t**, but at the combined action of the chaperone and Arg the *R*_h_*** value decreases.

Determination of the initial stoichiometry of the 14-3-3ζ_m_–UV-Ph*b* complex (*S*_0_) showed that in the absence of Arg the *S*_0_ value was equal to 30.2 ± 1.3 (*R*^2^ = 0.969; Figure 4A, curve 1) and in the presence of 0.1 M Arg *S*_0_ = 24.3 ± 0.4 (*R*^2^ = 0.995; Figure 4A, curve 2). It was taken into account that the molecular weight of 14-3-3ζ_m_ is equal to 28 kDa. As can be seen from Figure 4B, the relative duration of the nucleation stage, *t**/*t**_0_, increases with increasing [14-3-3ζ_m_]/[UV-Ph*b*] ratio both in the case of the absence and in the presence of Arg (curves 1 and 2, respectively).

### 2.2. The Effect of Arg on Tryptophan Fluorescence Spectra of Chaperones

To interpret the obtained data correctly, it was necessary to analyze how Arg affects the molecular structure of protein chaperones. For this purpose, the intrinsic tryptophan fluorescence of HspB6 and 14-3-3ζ_m_ was investigated in the presence of various concentrations of Arg at an excitation wavelength of 292 nm and a fixed value of ionic strength (0.15 M). Figure 5A shows the tryptophan fluorescence spectra of HspB6 (0.32 mg/mL) at 25 °C in the presence of Arg. The fluorescence spectroscopy is very sensitive to the local environment of tryptophan residues, and any changes in this area lead to a change in the tryptophan fluorescence intensity and the position of the Trp emission maximum (λ_max_). In order to avoid errors in determining the values of λ_max_, the obtained spectra were fitted using the following Equation (3) [42,43]:(3)$$F\left(\lambda \right)=\left\{\begin{array}{ll} F_{\max}\exp\left(-\frac{ln2}{{ln}^{2}p}{ln}^{2}\frac{a-\lambda }{a-\lambda_{\max}}\right) & \mathrm{when}\;\;\;\lambda <a\\ 0 & \mathrm{when}\;\;\;\lambda \geq a \end{array}\right.$$
where *F*_max_ is the fluorescence intensity maximum, λ_max_ is the position of the Trp emission maximum, *p* is an asymmetry parameter and *a* is the function limiting point.

An assessment of an area under the spectrum showed that an increase in Arg concentration from 0 to 0.02 M leads to the 35% growth of tryptophan fluorescence intensity of HspB6 (Figure 5A). A further increase in the concentration of Arg up to 0.09 M has only a slight effect on the intensity of tryptophan fluorescence. The dependence of the tryptophan fluorescence intensity on the ligand concentration at a fixed wavelength makes it possible to estimate the saturation of ligand-binding sites with Arg. A similar dependence is presented in Figure 5B, where the value of 338.8 nm corresponding to the position of λ_max_ for HspB6 without Arg is taken as a fixed wavelength. The values of fluorescence emission maximum at [Arg]→∞, *F*_lim_, and the dissociation constants, *K*_diss_, for complexes of Arg with HspB6 or 14-3-3ζ_m_ were calculated using Equation (4) [44]:(4)$$F=F_{0}+\frac{\left(F_{\lim}-F_{0}\right){\left[L\right]}_{0}}{\left(K_{\mathrm{diss}}+{\left[L\right]}_{0}\right)}$$
where *F*_0_ is the value of fluorescence emission maximum in the absence of Arg and [L]_0_ is concentration of Arg.

The value of *K*_diss_ for Arg–HspB6 complex was found to be equal to 0.010 ± 0.003 M and the value of *F*_lim_/*F*_0_ was 1.45 ± 0.04 (*R*^2^ = 0.971). This means that half-saturation of the ligand-binding sites of the HspB6 molecule with Arg occurs at a concentration [Arg]_0.5_ = 0.01 M, and the fluorescence intensity increases by 45% when all sites are completely saturated with Arg. It should be noted that Arg with a concentration of 0.09 M saturates the ligand-binding sites of the HspB6 molecule by almost 80%.

Another characteristic of the tryptophan fluorescence spectrum is the position of the Trp emission maximum (λ_max_). The position of λ_max_ for HspB6 without Arg corresponds to the wavelength of 338.8 ± 0.1 nm (Figure 5C). The addition of 0.005 M Arg leads to a shift of emission maximum λ_max_ to the value 339.5 ± 0.1 nm (*R*^2^ = 0.998). The further increase in the Arg concentration up to 0.09 M is accompanied by a slight shift in the maximum of fluorescence spectrum to longer wavelengths (red shift) up to the value λ_max_ = 340.1 ± 0.1 nm (*R*^2^ = 0.998). The red shift of λ_max_ position at Arg concentration corresponding to half-saturation of the ligand-binding sites of the HspB6 molecule ([Arg] = 0.01 M) was only 0.7 nm.

The more pronounced increase in tryptophan fluorescence intensity with the addition of Arg was observed for 14-3-3ζ_m_ (0.48 mg/mL, Figure 6). The area under the spectrum grew for 2.5 times with increasing Arg concentration up to 0.09 M (Figure 6A). The value of *K*_diss_ for Arg–14-3-3ζ_m_ complex calculated from the dependence of fluorescence intensity on [Arg] at λ = 336.5 nm (Figure 6B) was found to be equal to 0.049 ± 0.006 M (*R*^2^ = 0.994). Complete saturation of ligand-binding sites of the 14-3-3ζ_m_ molecule will lead to an increase in the fluorescence intensity by 3.3 times (*F*_lim_/*F*_0_ = 3.30 ± 0.09, *R*^2^ = 0.994), and 0.09 M Arg saturates these sites only by 64%. The position of the maximum of tryptophan fluorescence spectrum for 14-3-3ζ_m_, λ_max_, is equal to 336.5 ± 0.1 nm (*R*^2^ = 0.997) (Figure 6C). The addition of Arg in different concentrations from 0.005 to 0.09 M leads to a significant red shift of the emission maximum λ_max_ to an average value 340.1 ± 0.3 nm (*R*^2^ = 0.998) which was not statistically dependent on the concentration of Arg in solution. So the red shift of λ_max_ position for 14-3-3ζ_m_ in the presence of half-saturation concentration of Arg ([Arg] = 0.049 M) was 3.6 nm.

The values of the fluorescence spectra maxima, λ_max_, for HspB6 and 14-3-3ζ_m_ indicate that the location of tryptophan residues in protein chaperones is favorable for the formation of hydrogen bonds with the indole group of tryptophan. In the presence of Arg we observe an increase in the fluorescence intensity and a red shift of λ_max_ towards longer wavelengths for both chaperones. This fact suggests that conformational changes in the chaperone molecule induced by binding of Arg, perhaps stimulating the formation of new bonds with the solvent or with other amino acid residues of the protein.

### 2.3. The Effect of Arg on Thermal Stability of Chaperones

The structural changes in the protein molecule significantly affect the process of its thermal denaturation. Therefore, of special interest was to study the effect of Arg on the thermal unfolding of protein chaperones using the method of differential scanning calorimetry (DSC). Figure 7 shows the DSC profiles for HspB6 (1.2 mg/mL, Figure 7A) and 14-3-3ζ_m_ (1 mg/mL, Figure 7B) in the absence and presence of 0.1 M Arg at constant ionic strength of 0.15 M.

According to DSC data, the position of the thermal transition maximum (*T*_max_) for HspB6 shifted by 1.8 °C towards lower temperatures upon the addition of 0.1 M Arg: from 60.7 ± 0.1 °C to 58.9 ± 0.1 °C (Figure 7A). 14-3-3ζ_m_ is less thermostable than HspB6, and its *T*_max_ = 50.7 ± 0.1 °C (Figure 7B, curve 1). In the presence of 0.1 M Arg, the value of *T*_max_ for 14-3-3ζ_m_ shifts by 1.6 degrees towards the lower temperatures and is was found to be equal to 49.1 ± 0.1 °C (Figure 7B, curve 2). It should be noted that the calorimetric enthalpy (Δ*H*_cal_) of the thermal transitions, defined as the area under the DSC curve, does not change significantly upon addition of Arg for each of the chaperones under study. The main calorimetric parameters of the thermal unfolding of these proteins are presented in Table 2.

Thus, the slight shift of DSC profiles of HspB6 and 14-3-3ζ_m_ towards lower temperatures in the presence of Arg indicates a decrease in the thermal stability of protein chaperones. The binding of Arg to chaperone molecules is accompanied by conformational changes in their structures, which leads to the destabilization of these proteins.

## 3. Discussion

When comparing the effect of various additives on the anti-aggregation activity of protein chaperones, it is very important to consider that Arg can cause changes in the structure of target proteins, affecting their stability and aggregation pathway. It was shown that at high ionic strength (0.5 M), Arg stabilizes the UV-Ph*b* molecule with no effect on its aggregation pathway [36]. However, at physiological values of ionic strength (0.15 M), Arg causes changes in the structure of UV-Ph*b*, which result in its transformation into a more destabilized state and acceleration of the process of UV-Ph*b* aggregation at 37 °C [36]. On this test system we previously showed [40] that the rate of the aggregate growth (*v*_0_) was determined by the rate of the structural reorganization of UV-Ph*b* (stage P^0^→P^a^ in the scheme represented in Figure 8). 

Heating of UV-Ph*b* containing concealed damages (P^0^) at 37 °C results in the structural reorganization of UV-irradiated protein to the state with manifested damages (P^a^) (stage 1). Slow transformation of P^0^ into P^a^ is followed by nucleation (stage 2) and fast attachment of P^a^ to the existing aggregates with the formation of amorphous aggregates (stage 3).

The parameter *t** characterizes the duration of the nucleation stage (stage 2; Figure 8). Since Arg at ionic strength of 0.15 M decreases the duration of the nucleation stage of UV-Ph*b* [36], in order to compare the action of chaperones in the absence and in the presence of Arg, we constructed dependences of the relative duration of this stage (*t**/*t**_0_) on the [chaperone]/[UV-Ph*b*] ratio (Figure 2B and Figure 4B) which gives us the opportunity to make an assumption about the mechanism of the interaction of UV-Ph*b* with the chaperone in the presence of Arg. With increasing of HspB6 concentration, the *t**/*t**_0_ value increases in the presence of Arg, although it practically does not change in the absence of Arg (Figure 2B). For 14-3-3ζ_m_, the relative duration of the nucleation stage increases both in the case of addition and in the absence of Arg (Figure 4B), but in the presence of Arg this effect is more pronounced (curve 2 in Figure 4B). Thus, Arg stimulates the anti-aggregation activity of HspB6 and 14-3-3ζ_m_ already at the nucleation stage.

The size of the nuclei (*R*_h_*) at the end of the nucleation stage of UV-Ph*b* aggregation is 40 ± 2 nm and changes in the presence of various additives, as can be clearly seen from Table 1. The size of the UV-Ph*b* nuclei increases with the addition of 0.1 M Arg. The fact that aggregates of larger size are formed in the presence of Arg has been shown for aggregation of UV-Ph*b* at 37 °C and an ionic strength of 0.15 M [36]. It was suggested that in this case the formation of larger aggregates and, as a result, acceleration of UV-Ph*b* aggregation is due to conformational destabilization of UV-Ph*b* molecules in the presence of Arg, which is due to the interaction of the guanidine group of arginine and acidic residues of the protein. Similar data were obtained for aggregation of bovine serum albumin (BSA) at 70 °C [45], for dithiothreitol-induced aggregation of BSA at 45 °C [46] and for aggregation of phosphorylase kinase from rabbit skeletal muscle in the presence of 0.1 mM EGTA at 37 °C [34]. When UV-Ph*b* interacts with HspB6 or 14-3-3ζ_m_, the value of *R*_h_* also increases (Table 1). It can be assumed that in this case the nuclei formed during the nucleation stage include both reorganized molecules of UV-Ph*b* and chaperone molecules.

In the case of the simultaneous presence of Arg and one of the proteins the *R*_h_* value for aggregates becomes less than the corresponding value measured in the presence of only Arg or any of the protein chaperones (Table 1). This can be explained by the fact that when interacting with Arg, both chaperones and target protein undergo conformational transitions. This, in turn, leads to the fact that fewer chaperone molecules bind to the target protein molecule. The decrease in *S*_0_ observed in this case (Figure 2A and Figure 4A) indicates an increase in the efficiency of suppressing UV-Ph*b* aggregation by such modified chaperones.

The determined stoichiometry *S*_0_ characterizes the binding of the chaperone to the form of P^0^ (Figure 8, stage 1). The calculation of the initial stoichiometry of the chaperone–target protein complex (*S*_0_) values, based on the assumption that the protein chaperone forms a tight complex with target protein, showed that for the completely prevention of one UV-Ph*b* subunit aggregation ~12 subunits of HspB6 are needed. At the same time, it is needed almost 2.6 times more 14-3-3ζ_m_ subunits (~30 subunits) to achieve the similar protective effect. It should be emphasized that similar data were obtained in the presence of 0.1 M Arg: the complete prevention of UV-Ph*b* aggregation requested almost three times more subunits of 14-3-3ζ_m_ than HspB6 (~8 and 24 subunits of HspB6 and 14-3-3ζ_m_, respectively). However, the most important result is a significant decrease in the *S*_0_ value in the presence of Arg observed for both proteins under study. For HspB6 the *S*_0_ value in the presence of 0.1 M Arg decreased by 1.4 times and for 14-3-3ζ_m_ 1.2-fold decrease in the *S*_0_ value took place. These data suggest that the anti-aggregation activity of these chaperones increases in the presence of 0.1 M Arg. The effect of Arg on the *S*_0_ value (Figure 2A and Figure 4A) is associated with Arg-induced changes in the conformational state of the protein chaperone and UV-Ph*b*, these changes resulting in alteration of the chaperone affinity for the target protein. The presented data indicate that for the test system, based on the thermal aggregation of UV-Ph*b* at 37 °C, the small heat shock protein HspB6 has a higher anti-aggregation activity than the monomeric form of 14-3-3ζ.

It is well known, that for the manifestation of protein chaperone activity and contacting with partially unfolded target proteins, the appearance of hydrophobic sites on the surface of chaperones is necessary [47]. It was demonstrated that the exposure of hydrophobic surfaces in chaperones was connected with increasing chaperone-like activity of Hsp16.3 from *Mycobacterium tuberculosis* [47], α-crystallin [48] and GroEL [49]. One of the agents leading to a change in the hydrophobicity of chaperones is Arg. As was demonstrated by Srinivas et al. [37], Arg increases hydrophobic surfaces in α-crystallin that results in an increase in its anti-aggregation activity. However, Arg does not always enhance anti-aggregation activity of chaperones. In case of catalase aggregation at 55 °C, Arg decreases the ability of αB-crystallin to inhibit aggregation of the target protein, but in case of fibril formation of reduced and carboxymethylated κ-casein, Arg has no effect on anti-aggregation activity of αB-crystallin [39].

However, up to now there were no data on the effect of Arg on HspB6 and 14-3-3ζ_m_ structures. To characterize the effect of Arg directly on protein chaperones, fluorescence spectroscopy and DSC methods were used. Changes in the intrinsic tryptophan fluorescence of proteins provide information on the conformational changes associated with binding of the ligand. On the basis of approaches elaborated by Burstein et al. [50] we made the conclusion that upon binding of Arg the tryptophan residues of HspB6 and 14-3-3ζ_m_ become more solvent exposed. The addition of Arg provokes conformational changes in the structures of these proteins, leading to greater accessibility of tryptophan residues to the external hydrophilic environment. Tsumoto et al. [51] suggested that Arg affected protein conformation through interaction of its guanidine moiety with tryptophan on the surface of the protein molecule. The fact that the interaction with Arg results in a change in the conformational state of HspB6 and 14-3-3ζ_m_ molecules is confirmed by DSC data. A decrease in the thermal stability of chaperones in the presence of 0.1 M Arg at ionic strength of 0.15 M indicates destabilization of the tertiary structures of these proteins.

The binding of Arg to the HspB6 molecule is stronger than to the molecule of 14-3-3ζ_m_: the dissociation constant for the Arg–HspB6 complex is 5 times less than the corresponding value for the Arg–14-3-3ζ_m_ complex. The concentration of Arg required for half-saturation of the ligand-binding sites of the HspB6 molecule is only 0.01 M, while for 14-3-3ζ_m_ this value is 0.049 M. Moreover, according to fluorescence spectroscopy, the changes in the tryptophan residues environment of the 14-3-3ζ_m_ molecule are significantly more pronounced than for HspB6. This can be explained by the difference in protein structures. According to Protein Data Bank, monomeric 14-3-3 contains 81% alpha-helices, which are less stable compared to beta-strands. On the other hand, HspB6 contains only 5% alpha-helices and 44% beta-strands, that may explain its resistance to the destabilizing effect of Arg.

In conclusion, it should be emphasized that changes in the conformational state and a decrease in thermal stability of the protein chaperones, as well as an increase in their anti-aggregation activity in the presence of Arg, are caused not by a change in the ionic strength of the solution, but directly by the effect of Arg on these proteins, because the ionic strength remained constant when carrying out all experiments. It is well known that various modulators can affect the functioning of protein chaperones by binding to the target protein or to the chaperones themselves. So, phosphorylation of HspB6 is important for the regulation of its functions in different type of muscles [52]. In addition, the oligomeric state of HspB6 changes during phosphorylation, which affects its interaction with other proteins, for example, HspB1 [12]. It is also known that phosphate can affect the interaction of 14-3-3 protein with various target proteins, in particular, inorganic phosphate induces dissociation of phosphorylated HspB6 complexes with 14-3-3γ, 14-3-3ζ and 14-3-3ζ_m_ [53]. In the present work it was shown that the presence of such a modulator as Arg can lead to the destabilization of proteins, both of the target protein and the protein chaperones interacting with it. It’s interesting that the anti-aggregation activity of the chaperones in this case can increase, as it was shown by the example of HspB6 and 14-3-3ζ_m_. This fact should be taken into account when studying the effect of Arg on various systems and complexes.

## 4. Materials and Methods

### 4.1. Materials

Hepes and L-arginine monohydrochloride were purchased from “Sigma-Aldrich” (St. Louis, MO, USA), 1,4-dithiothreitol (DTT) was purchased from PanReac (Barcelona, Spain), NaCl was purchased from “Reakhim” (Moscow, Russia). The procedure of isolation of Ph*b* from rabbit skeletal muscles was described in [54].

### 4.2. Isolation of Chaperones

Recombinant human wild type HspB6 (Hsp20) (Uniprot ID O14558) were expressed and purified as described in [16], human untagged 14-3-3ζ wild type protein (Uniprot ID P63104) and its monomeric mutant form 14-3-3ζ_m_ were expressed and purified as described previously [16,25].

### 4.3. Light Scattering Intensity Measurements

The procedure UV-irradiation of Ph*b* was described in [40]. The kinetics of UV-Ph*b* thermal aggregation at 37 °C in the absence of any additives and in the presence of protein chaperones alone or together with 0.1 M Arg was studied in 0.03 M Hepes buffer, pH 6.8, with 0.15 M NaCl and 0.5 mM DTT at constant ionic strength equal to 0.15 M. When studying the effect of Arg, concentration of NaCl in the buffer was varied so that the final ionic strength was constant and equal to 0.15 M. The light scattering intensity of the samples was recorded with a commercial instrument Photocor Complex (PhotoCor Instruments Inc., College Park, MD, USA) with a He-Ne laser (Coherent, Santa Clara, CA, USA, Model 31-2082, 632.8 nm, 10 mW) as a light source. To study the effects of protein chaperones in the absence or in the presence of 0.1 M Arg on UV-Ph*b* aggregation, the solutions of protein chaperones in buffer were preincubated for 5 min at 37 °C in the cell before the addition of the UV-Ph*b*. The aggregation process was initiated by the addition of UV-Ph*b* to the final volume of 0.45 mL. The scattered light was collected at a 90° angle. DynaLS software (Alango, Tirat Carmel, Israel) was employed for polydisperse analysis of dynamic light scattering (DLS) data [36,40,55].

### 4.4. Tryptophan Fluorescence of Chaperones

Tryptophan fluorescence spectra of HspB6 and 14-3-3ζ_m_ in 0.03 M Hepes buffer, pH 6.8, containing 0.15 M NaCl and 0.5 mM DTT in the absence or in the presence of 0.005–0.09 M Arg at constant ionic strength (0.15 M) were recorded using a spectrofluorophotometer RF-5301 PC (Shimadzu, Kyoto, Japan) in 0.3 cm path length cuvette. Fluorescence measurements were carried out at the temperature of 25 °C. The optical density of the samples at 280 nm was 0.1–0.2. The excitation wavelength was 292 nm, and fluorescence emission was monitored between 300 and 450 nm. The excitation and emission spectral slit widths were 2 nm. Spectra of blank sample were subtracted from the main spectra.

### 4.5. Thermal Denaturation of Chaperones

Differential scanning calorimetry was used to investigate the thermal denaturation of 14-3-3ζ_m_ and HspB6 in the absence or in the presence of 0.1 M Arg. The experiments were performed on a MicroCal VP-Capillary DSC differential scanning calorimeter (Malvern Instruments, Northampton, MA 01060, USA) at a heating rate of 1 °C/min in 0.03 M Hepes buffer, pH 6.8. The ionic strength in all experiments was 0.15 M. The protein concentration was 1 mg/mL for the 14-3-3ζ_m_ and 1.2 mg/mL for HspB6. In the experiments with 0.1 M Arg, the same concentration of Arg was added to the both control and sample cells. The correction of the calorimetric traces, analysis of the temperature dependence of excess heat capacity, the thermal stability estimation and calculation of calorimetric enthalpy (Δ*H*_cal_) was performed as described in [56].

## Figures and Tables

**Figure 1 ijms-21-02039-f001:**
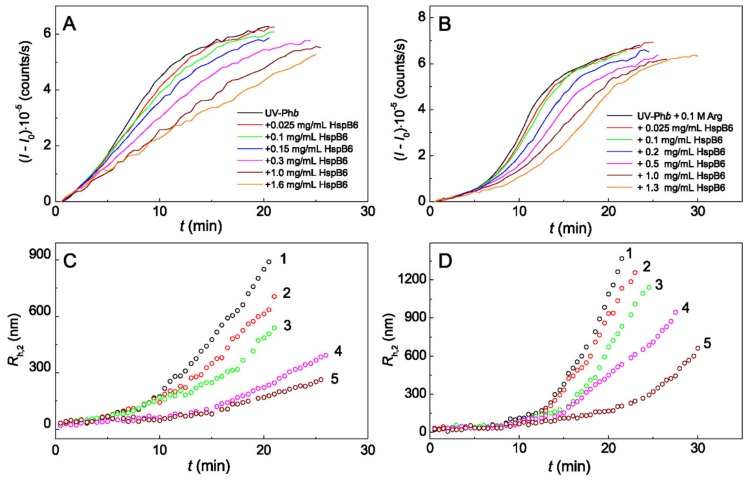
Effect of HspB6 on the kinetics of aggregation of UV-Ph*b* (0.25 mg/mL) in the absence (**A**,**C**) and in the presence of 0.1 M Arg (**B**,**D**) at 37 °C. (**A**,**B**) The dependences of the light scattering intensity (*I* − *I*_0_) on time. The concentrations of HspB6 are shown in panels (**A**,**B**). (**C**,**D**) The dependences of the hydrodynamic radius *R*_h,2_ of protein aggregates on time obtained at the following concentrations of HspB6: (1) 0, (2) 0.025, (3) 0.1, (4) 0.3 and (5) 1 mg/mL.

**Figure 2 ijms-21-02039-f002:**
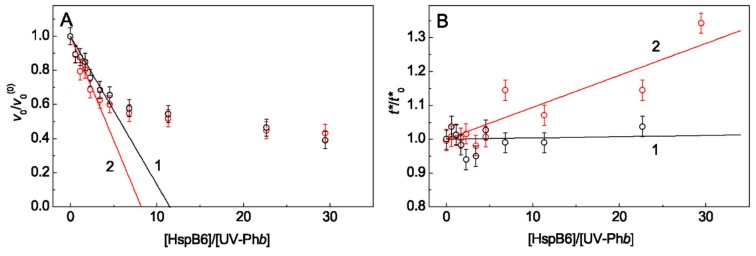
Effect of HspB6 on the kinetic parameters of UV-Ph*b* (0.25 mg/mL) aggregation in the absence and in the presence of 0.1 M Arg at 37 °C. (**A**) The dependences of the relative initial rate of UV-Ph*b* aggregation for the stage of aggregate growth, *v*_0/_*v*_0_^(0)^, and (**B**) the relative value of the nucleation stage duration, *t**/*t**_0_, on the ratio of molar concentrations of HspB6 and UV-Ph*b*. Points are experimental data. The solid lines in panel A are calculated from Equation (2). Curves 1 and 2 in panels (**A**,**B**) are obtained in the absence and in the presence of 0.1 M Arg, respectively.

**Figure 3 ijms-21-02039-f003:**
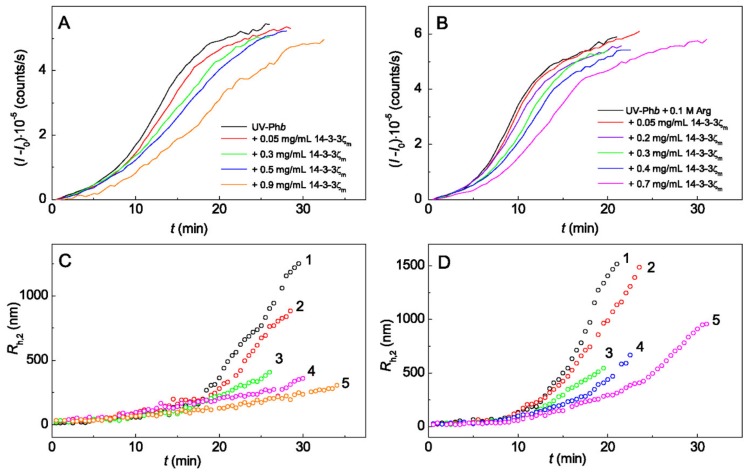
Effect of 14-3-3ζ_m_ on the kinetics of aggregation of UV-Ph*b* (0.25 mg/mL) in the absence (**A**,**C**) and in the presence of 0.1 M Arg (**B**,**D**) at 37 °C. (**A**,**B**) The dependences of the light scattering intensity (*I* − *I*_0_) on time. The concentrations of 14-3-3ζ_m_ are shown in panels (**A**,**B**). (**C**,**D**) The dependences of the hydrodynamic radius *R*_h,2_ of protein aggregates on time obtained at the following concentrations of 14-3-3ζ_m_: in panel (**C**) (1) 0, (2) 0.05, (3) 0.3, (4) 0.7 and (5) 0.9 mg/mL; in panel (**D**) (1) 0, (2) 0.05, (3) 0.3, (4) 0.4, and (5) 0.7 mg/mL.

**Figure 4 ijms-21-02039-f004:**
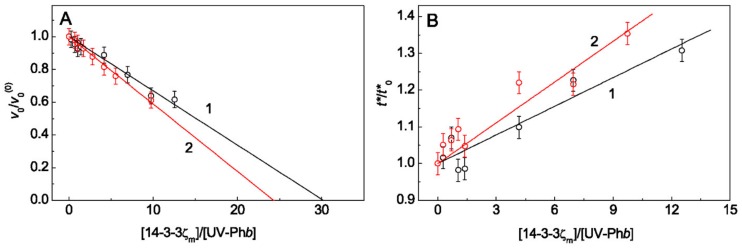
Effect of 14-3-3ζ_m_ on the kinetic parameters of UV-Ph*b* (0.25 mg/mL) aggregation in the absence and in the presence of 0.1 M Arg at 37 °C. (**A**) The dependences of the relative initial rate of UV-Ph*b* aggregation at the stage of aggregate growth, *v*_0_/ *v*_0_^(0)^, and (**B**) the relative value of the nucleation stage duration, *t**/*t**_0_, on the ratio of molar concentrations of 14-3-3ζ_m_ and UV-Ph*b*. Points are experimental data. The solid lines in panel (**A**) are calculated from Equation (2). Curves 1 and 2 in panels (**A**,**B**) are obtained in the absence and in the presence of 0.1 M Arg, respectively.

**Figure 5 ijms-21-02039-f005:**
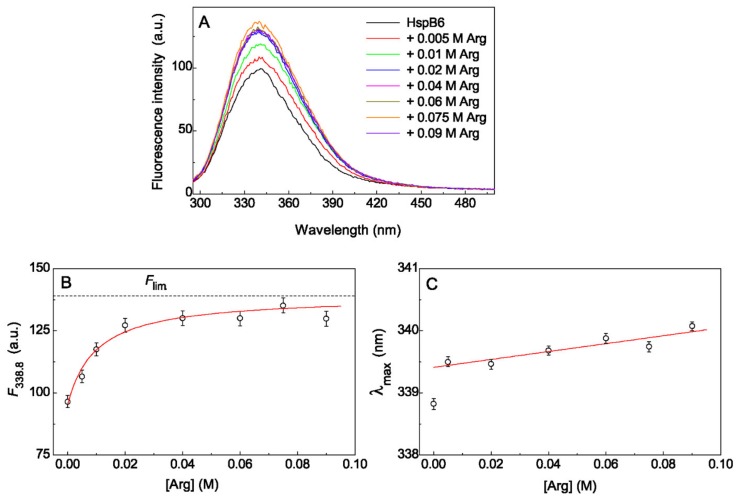
The effect of Arg on tryptophan fluorescence of HspB6. (**A**) The tryptophan fluorescence spectra of HspB6 (0.32 mg/mL) in 0.03 M Hepes-buffer, containing 0.5 mM dithiothreitol (DTT), and in the presence of increasing concentrations of Arg from 0 to 0.09 M. The spectra were obtained at 25 °C, excitation wavelength of 292 nm and constant ionic strength of 0.15 M. (**B**) The dependence of fluorescence intensity on Arg concentration at λ = 338.8 nm. Points are the experimental data. The solid curve was calculated using Equation (4) at *K*_diss_ = 0.01 M, *F*_0_ = 95.6 and *F*_lim_ = 139. The horizontal dash line corresponds to *F*_lim_ value. (**C**) The dependence of Trp emission maximum (λ_max_) on Arg concentration.

**Figure 6 ijms-21-02039-f006:**
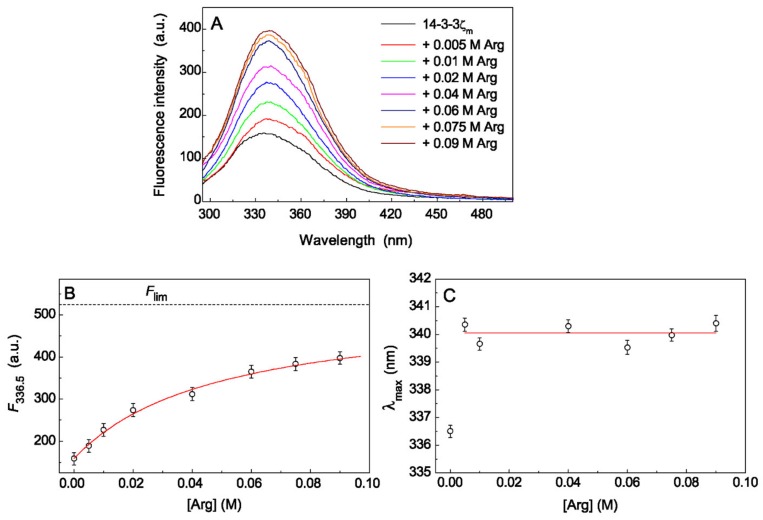
The effect of Arg on tryptophan fluorescence of 14-3-3ζ_m_. (**A**) The tryptophan fluorescence spectra of 14-3-3ζ_m_ (0.48 mg/mL) in 0.03 M Hepes-buffer, containing 0.5 mM DTT, and in the presence of increasing concentrations of Arg from 0 to 0.09 M. The spectra were obtained at 25 °C, excitation wavelength of 292 nm and constant ionic strength of 0.15 M. (**B**) The dependence of fluorescence intensity on Arg concentration at λ = 336.5 nm. Points are the experimental data. The solid curve was calculated using Equation (4) at *K*_diss_ = 0.049 M, *F*_0_ = 159 and *F*_lim_ = 524. The horizontal dash line corresponds to *F*_lim_ value. (**C**) The dependence of Trp emission maximum (λ_max_) on Arg concentration.

**Figure 7 ijms-21-02039-f007:**
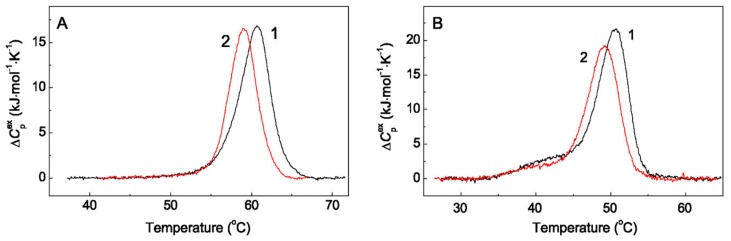
The effect of Arg on thermal stability of protein chaperones. The temperature dependences of the excess heat capacity (*C*_p_) obtained by DSC (**A**) for HspB6 (1.2 mg/mL) and (**B**) for 14-3-3ζ_m_ (1 mg/mL) in the absence and in the presence of 0.1 M Arg (curves 1 and 2, respectively). The ionic strength in both cases was 0.15 M. The heating rate was 1 °C/min.

**Figure 8 ijms-21-02039-f008:**
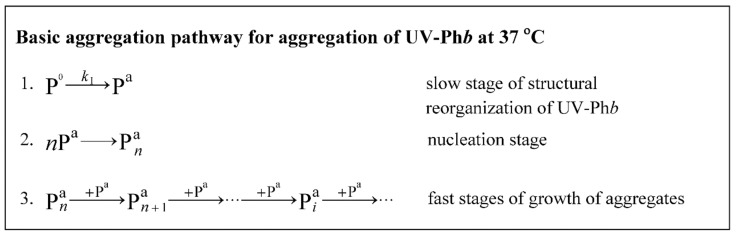
Scheme illustrating basic aggregation pathway for UV-Ph*b* aggregation at 37 °C [40].

**Table 1 ijms-21-02039-t001:** The values of *R*_h_* for UV-Ph*b* aggregates in the absence and in the presence of protein chaperones and 0.1 M Arg.

Additives	*R*_h_* (nm)
No additives	40 ± 2
+ Arg 0.1 M	49 ± 2
+ HspB6	49 ± 2
+ Arg 0.1 M + HspB6	46 ± 2
+ 14-3-3ζ_m_	59 ± 6
+ Arg 0.1 M + 14-3-3ζ_m_	44 ± 5

**Table 2 ijms-21-02039-t002:** Calorimetric parameters obtained from the DSC data for the thermal transitions of protein chaperones in the absence and in the presence of 0.1 M Arg.

Sample	*T*_max_, °C	Δ*H*_cal_, kJ·mol^−1^
HspB6	60.7 ± 0.1	86 ± 8
HspB6 + 0.1 M Arg	58.9 ± 0.1	75 ± 7
14-3-3ζ_m_	50.7 ± 0.1	136 ± 12
14-3-3ζ_m_ + 0.1 M Arg	49.1 ± 0.1	120 ± 11

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
