# Peer review of "Effect of Arginine on Chaperone-Like Activity of HspB6 and Monomeric 14-3-3ζ"

_ijms, 2020, doi:10.3390/ijms21062039_

Round 1

Reviewer 1 Report

This is a very interesting manuscript on the important topic of modulation of protein aggregation by two important protein chaperones. Since protein aggregation is at the basis of important human disease with the process so far not amenable to pharmacologic intervention, a study like this will make a lot of impact in the field.

To strengthen the latter aspect, it might be very helpful to discuss the potential for such a small-molecule modulation of both protein classes. If I am not mistaken, for both HspB6 and 14-3-3 a number of small molecule binders and PPI modulators is described and at least in the case of 14-3-3, detailed structural information is available. Viewing the results of this manuscript in the light of potential intervention would significantly increase the interest and impact of the story. In addition, such molecules should be tested with the methodology described here. While clearly out of scope for the current manuscript, the results of such experiments will make great material for follow-up studies.

Author Response

Dear Reviewer,

Thank you very much for your revision concerning our paper “Effect of Arginine on Chaperone-like Activity of HspB6 and Monomeric 14-3-3ζm”. We are thankful to you for the valuable comments. We took them into account by correcting our article.

Summary

This is a very interesting manuscript on the important topic of modulation of protein aggregation by two important protein chaperones. Since protein aggregation is at the basis of important human disease with the process so far not amenable to pharmacologic intervention, a study like this will make a lot of impact in the field.

Comments.

To strengthen the latter aspect, it might be very helpful to discuss the potential for such a small-molecule modulation of both protein classes. If I am not mistaken, for both HspB6 and 14-3-3 a number of small molecule binders and PPI modulators is described and at least in the case of 14-3-3, detailed structural information is available. Viewing the results of this manuscript in the light of potential intervention would significantly increase the interest and impact of the story. In addition, such molecules should be tested with the methodology described here. While clearly out of scope for the current manuscript, the results of such experiments will make great material for follow-up studies.

Reply

Discussion, page 12, lines 399-410. The next phrases have been inserted: “It is well known that various modulators can influence the anti-aggregation activity of protein chaperones by binding to the target protein or to the chaperones themselves. So phosphorylation of HspB6 is important for the regulation of its functions in different type of muscles [53]. In addition, the oligomeric state of HspB6 changes during phosphorylation, which affects its interaction with other proteins, for example, HspB1 [12]. It is also known that phosphate can affect the interaction of 14-3-3 protein with various target proteins, in particular, inorganic phosphate induces dissociation of phosphorylated HspB6 complexes with 14-3-3g, 14-3-3z and 14-3-3zm [54]. In the present work it was shown that the presence of such a modulator as Arg can lead to the destabilization of proteins, both of the target protein and the protein chaperones interacting with it. It’s interesting that the anti-aggregation activity of the chaperones in this case can increase, as it was shown by the example of HspB6 and 14-3-3zm. This fact should be taken into account when studying the effect of Arg on various systems and complexes.”

New references have been inserted:

Lines 651-656

  1. Dreiza, C.M.; Komalavilas, P.; Furnish, E.J.; Flynn C.R.; Sheller M.R.; Smoke C.C.; Lopes L.B.; Brophy C.M. The small heat shock protein, HSPB6, in muscle function and disease. Cell Stress Chaperones 2010, 15, 1–11. https://doi.org/10.1007/s12192-009-0127-8.
  2. Sluchanko, N.N.; Chebotareva, N.A.; Gusev N.B. Modulation of 14-3-3/phosphotarget interaction by physiological concentrations of phosphate and glycerophosphates. PLoS ONE 2013, 8, e72597. https://doi:10.1371/journal.pone.0072597.

Reviewer 2 Report

In the article by Mikhayalova et al, the authors explore the anti-aggregation effects of the HspB6 and 14-3-3 ζ in the presence of a chemical chaperone, arginine. Molecular chaperones have been highly studied for many years, yet their mechanisms still aren’t fully understood. This work is of great interest to the field and could have potential impacts for other related small heat shock proteins. Through complementary methods, the authors show that Arg in combination with HspB6 or 14-3-3 ζ is able to suppress aggregation activity by promoting a conformational change in each chaperone.  Overall, I found the article to be very thorough and well written. I have no major concerns; however, I have a few minor questions and suggestions.

It wasn’t clear in the introduction why these two particular chaperones were chosen for this study and whether they are related. It appears that the two chaperones also collaborate and interact in vivo. Is this collaboration relevant for anti-aggregation activity and show any stimulation of the anti-aggregation effect in the absence or presence of Arg?

In Figure 1 – Is complete suppression of aggregation possible with the chaperones alone or chaperones + Arg or are they only able to slow the rate of aggregation? Also, there is mention that UV-Phb aggregation can form both large and small aggregates but this study only looks at the large aggregates. Is there any detectable effect on the small aggregates?

The authors show that 0.1M concentration of Arg is responsible for increasing the anti-aggregation activity of the chaperones. How does this relate to concentrations of free arginine in the cell? It is also stated that Arg binds and stabilizes UV-Phb without changing the aggregation pathway; this work also demonstrates that Arg binds to the chaperones. Is there any indication of the Arg binding site or substrate binding site on the chaperones and if so, do they overlap? Is it possible to tease out whether Arg binding to both the chaperones and UV-Phb is required for stimulation of anti-aggregation activity or whether it is only the binding to the chaperone that is responsible for this effect?

Author Response

Dear Reviewer,

Thank you very much for your revision concerning our paper “Effect of Arginine on Chaperone-like Activity of HspB6 and Monomeric 14-3-3ζm”. We are thankful to you for the valuable comments. We took them into account by correcting our article.

Summary

In the article by Mikhaylova et al, the authors explore the anti-aggregation effects of the HspB6 and 14-3-3 ζ in the presence of a chemical chaperone, arginine. Molecular chaperones have been highly studied for many years, yet their mechanisms still aren’t fully understood. This work is of great interest to the field and could have potential impacts for other related small heat shock proteins. Through complementary methods, the authors show that Arg in combination with HspB6 or 14-3-3 ζ is able to suppress aggregation activity by promoting a conformational change in each chaperone. Overall, I found the article to be very thorough and well written. I have no major concerns; however, I have a few minor questions and suggestions.

Comments

Comment 1. It wasn’t clear in the introduction why these two particular chaperones were chosen for this study and whether they are related. It appears that the two chaperones also collaborate and interact in vivo. Is this collaboration relevant for anti-aggregation activity and show any stimulation of the anti-aggregation effect in the absence or presence of Arg?

Reply

It has been shown in vitro that 14-3-3ζ interacted with phosphorylated HspB6 and this interaction can be modulated by physiological concentrations of phosphate and glycerophospate [Sluchanko, Chebotareva, Gusev (2013), PlosOne 8, e72597]. We can suggest that the two chaperones also collaborate and interact in vivo.

As for the effect of Arg on the anti-aggregation activity of the 14-3-3/phosphorylated HspB6 complex, unfortunately, such experiments were not carried out due to technical reasons for the lack of sufficient proteins. This question is undoubtedly of scientific interest and we hope to get an answer to it in the course of further work.

The next changes have been made to the article:

Introduction, page 2, line 50, the next phrase: "(for example, it forms heterodimers with HspB6)," has been inserted;Introduction, page 2, lines 53-61, phrases: "them possess chaperone-like activity. It was shown using different target proteins that monomeric form of 14-3-3ζ might have comparable or even higher anti-aggregation activity than either the dimeric form of 14-3-3ζ or HspB6 and HspB5 [24,25]" have been omitted. Instead of it the following phrases: "of isoforms (e and g) are dimeric, whereas other isoforms exist as a mixture of dimer/monomer [23]. It was reported that phosphorylation of Ser58 of certain 14-3-3 isoforms (h, β, and z) in response to activation of different pathways could induce partial dissociation of 14-3-3 dimer (see a review [22]). The structure and functions of the 14-3-3 dimer are well characterized, while much less is known about the properties of the monomeric form of 14-3-3. Earlier it was shown that monomeric 14-3-3ζ has a chaperone-like activity and is stabilized by phosphorylated HspB6 [24]. Previously using different target proteins we found that monomeric form of 14-3-3ζ might have comparable or even higher chaperone-like activity than either the dimeric form of 14-3-3ζ or HspB6 and HspB5 [25]." have been inserted.

Comment 2. In Figure 1 – Is complete suppression of aggregation possible with the chaperones alone or chaperones + Arg or are they only able to slow the rate of aggregation? Also, there is mention that UV-Phb aggregation can form both large and small aggregates but this study only looks at the large aggregates. Is there any detectable effect on the small aggregates?

Reply

Suppression of UV-Phb aggregation occurs both in the presence of a single chaperone and a chaperone + Arg. However, in the presence of Arg, the anti-aggregation effect of chaperones is more pronounced: in the presence of Arg, the rate of UV- Phb aggregates growth, characterized by the parameter v0, decreases for both chaperones compared to the situation without Arg, and the duration of the nucleation stage, characterized by the parameter t*, on the contrary, increases. It is interesting to note that in the absence of Arg, the chaperone HspB6 practically does not affect the nucleation stage.

As for the formation of aggregates with small and large hydrodynamic radii (Rh,1 and Rh,2), as was mentioned in the article, large radii make the main contribution to the light scattering intensity detected by the DLS method. Small aggregates do not contribute significantly to this process. That is why, in this work, only the values of Rh,2 are presented.

Comment 3. The authors show that 0.1 M concentration of Arg is responsible for increasing the anti-aggregation activity of the chaperones. How does this relate to concentrations of free arginine in the cell? It is also stated that Arg binds and stabilizes UV-Phb without changing the aggregation pathway; this work also demonstrates that Arg binds to the chaperones. Is there any indication of the Arg binding site or substrate binding site on the chaperones and if so, do they overlap? Is it possible to tease out whether Arg binding to both the chaperones and UV-Phb is required for stimulation of anti-aggregation activity or whether it is only the binding to the chaperone that is responsible for this effect?

Reply

"How does this relate to concentrations of free arginine in the cell?"

Arg belongs to the category of Amino acids, which are produced in the human body only under certain conditions. With the slightest pathology, the human body does not have time to cope with the production of Arg. At rest, fasting serum L-arginine levels (without additives) are 71 ± 4 nmol/L or 15.1 ± 2.6 mg/mL. Oral administration of an arginine supplement with a dosage of 5 g on an empty stomach promotes an increase in AUC L-arginine (within 5 hours) by 64%. It is believed that in the healthy person should be from 50 to 150 μmoles of Arg. And the values of the Kdis for the 14-3-3ζm and HspB6 are equal to 49 mM, and 10 mM, respectively. But according to the standards developed by sanitary doctors, the daily requirement for Arg consumption is from 4.0 g (19 mM) for children, to 6.0 g (28.5 mM) for an adult.

"It is also stated that Arg binds and stabilizes UV-Phb without changing the aggregation pathway"

In our previous article [reference 36: Int. J. Biol. Macromol. 2018, 118, 1193–1202], we have showed that in the presence of Arg, the process of UV-Phb aggregation at 37 oC and ionic strength of 0.15 M passes through the formation of larger aggregates, and accordingly, the aggregation pathway changes.

"this work also demonstrates that Arg binds to the chaperones "

As indicated in the Discussion, Arg binds to both chaperone and UV-Phb. It is well known that the substrate largely determines whether chaperone will inhibit aggregation or accelerate it. In this case, Arg destabilizes both the chaperones and the target protein. As a result of such destabilization of proteins, we observe that fewer chaperone molecules are required in order to prevent aggregation of the molecule of target protein. It is likely that this mechanism can work on other substrates.

"Is there any indication of the Arg binding site or substrate binding site on the chaperones and if so, do they overlap?"

A decrease in the S0 value in the presence of Arg (Fig. 2A, Fig. 4A) is possible if there is no overlap of the Arg binding site and the substrate binding site on the chaperones. Otherwise, Arg would interfere with chaperone binding to UV-Phb and an increase in the S0 value would be observed.

"Is it possible to tease out whether Arg binding to both the chaperones and UV-Phb is required for stimulation of anti-aggregation activity or whether it is only the binding to the chaperone that is responsible for this effect?"

In the figures Fig. 2A, Fig. 4A, the relative value v0/v0(0) is plotted on the ordinate axis. This means that these figures demonstrate the effect of Arg precisely as a result of its binding on the chaperone.

Results, page 4, lines 145-147, the next phrase has been inserted: "The fact that the relative value v0/v0(0) is plotted on the ordinate axis means that the obtained dependence demonstrates the Arg effect precisely as a result of its binding directly to the chaperone."

Reviewer 3 Report

Manuscript ID: ijms-738580

Title: Effect of Arginine on Chaperone-like Activity of HspB6 and Monomeric 14-3-3

Summary

The authors have set out to determine the effects of arginine on the chaperone functions of two chaperones, HspB6 and 14-3-3. Using a combination of dynamic light scattering, tryptophan fluorescence, and differential scanning calorimetry methods, the authors have clearly demonstrated that arginine impacts the kinetics of aggregation for a model protein substrate, and, that arginine also destabilizes both chaperones. Of note, dynamic light scattering experiments have allowed for clear estimation of the chaperone:UV-Phb stoichiometry for the active complex. This should clarify some ambiguities regarding chaperone oligomerization discussed in the introduction. Overall, this manuscript is well-written and serves as a good example of the strength of traditional biophysical methods in addressing fundamental questions of enzyme function. However, there are certain instances (described below) where the authors may over-interpret their data. The generation of homology models or using existing structural data would rectify this issue though. All comments are summarized below.

Comments

  • 1, Lines 35-39: The authors state that HspB6 forms oligomers corresponding to dimers and tetramers, but then follow up with a statement suggesting this is somewhat odd because sHsps family chaperones tend to form large oligomers that easily undergo subunit exchange.
    • My questions here relate to what we are calling “normal.” Here, the authors do not define what “large oligomer” means. Is this also a tetramer, but simply one built from large molecular weight monomers? Or are we talking about dodecamers, hexamers, etc.? This information is necessary for the reader to draw valid comparisons. Moreover, the observation of dimers and tetramers for HspB6 suggests that subunit exchange can occur similar to the “large oligomers.”
  • 2, Lines 55-56: Please elaborate on this monomeric mutant. What has been done to promote disrupt oligomerization? Was this reported elsewhere or is it a novel result being reported here?
  • 3, Line 82: Please describe the experimental approach here. Concentrations, temperatures, etc. are provided. However, the thermal aggregation of UV-Phb is unclear. Was the protein previously UV-treated, but then aggregates at 37 degrees Celsius later? If so, is this when HspB6 is added? These details would aid greatly in reader comprehension of this experimental design.
  • 4, Line 85: “The initial part of the sigmoidal kinetic curve..” What curve? Please provided figure references if the reader is to follow what is written.
  • 5, Lines 104,105: “In addition, an increase in the chaperone concentration leads to a decrease in the value of the radii (Rh,2) both in the absence and in the presence of Arg.”
    • Comparison of Figures 1C and 1D reveal a clear lag phase prior to increase in the hydrodynamic radius. However, conditions lacking Arg seem to reveal slower kinetics. If the increase in Rh,2 as a function of time is due to UV-Phb aggregation, can the authors comment on these apparent kinetic differences? Why does the presence of arginine seem to accelerate aggregate formation?
    • This observation appears to quantified in Table 1. However, it is still unclear why the aggregate structure would have a lower hydrodynamic radius than the native (or remodeled) state would.
  • 6, Equation 3: Can the authors expand their explanation of this function? Though references are provided, it would make it easier for readers to follow if some explanation were provided. It is quite common to see tryptophan emissions spectra subjected to NLLS analyses, but usually parameters such as lmax or peak area are used, which are model-independent parameters. For this reason, I doubt most readers are going to be familiar with Eq. 3.
  • 7, Lines 244-245: The authors state that tryptophan residues are located on the surface of the protein in a hydrophilic environment. This statement is an overstatement based on the provided data. The data indicate that the relevant tryptophan residues are in an environment that favors hydrogen bonding to the indole group of tryptophan, but say nothing about solvent accessibility or location. Two equally valid models could be proposed: 1) tryptophan residues undergo an increase in solvent accessibility upon Arg binding or 2) Arg binds adjacent to a tryptophan and directly interacts. However, we have no way of distinguishing between these two models, which requires a more conservative interpretation.
  • 8, Lines 249-250: Again, this appears to be somewhat overstated. The conclusion is that the effect of Arg on the conformational state of 14-3-3 is more significant than on HspB6. The affinities suggest that binding to HspB6 is stronger. However, the change in fluorescence intensity is greater for 14-3-3 upon Arg binding. This is likely just a consequence of where tryptophan residues are located in overall protein structure. We must remember that the change in overall fluorescence intensity is directly linked to changes in local solvent properties for tryptophan residues. Therefore, we cannot say anything about impact on conformational state beyond the fact that a conformational change does likely occur.
  • 9, Figure 7: Have the authors performed DSC experiments +/- Arg for UV-Phb? It would be interesting to see if Arg also destabilizes this protein, which could make it easier for the chaperones to carry out their respective functions.
  • 10, Line 350: The authors could strengthen their argument for where tryptophan residues are located in overall structure by the generation of homology models that allow for prediction. Alternatively, mapping tryptophan locations onto structures of known homologs could work as well.
  • 11, Do the authors know what the concentration of arg is in living cells? Do the KD values reported here approach physiologic levels? This information could be helpful for readers in determining whether the behavior reported here may be relevant to in vivo

Author Response

Dear Reviewer,

Thank you very much for your revision concerning our paper “Effect of Arginine on Chaperone-like Activity of HspB6 and Monomeric 14-3-3ζm”. We are thankful to you for the valuable comments. We took them into account by correcting our article.

Summary

The authors have set out to determine the effects of arginine on the chaperone functions of two chaperones, HspB6 and 14-3-3. Using a combination of dynamic light scattering, tryptophan fluorescence, and differential scanning calorimetry methods, the authors have clearly demonstrated that arginine impacts the kinetics of aggregation for a model protein substrate, and, that arginine also destabilizes both chaperones. Of note, dynamic light scattering experiments have allowed for clear estimation of the chaperone: UV-Phb stoichiometry for the active complex. This should clarify some ambiguities regarding chaperone oligomerization discussed in the introduction. Overall, this manuscript is well-written and serves as a good example of the strength of traditional biophysical methods in addressing fundamental questions of enzyme function. However, there are certain instances (described below) where the authors may over-interpret their data. The generation of homology models or using existing structural data would rectify this issue though. All comments are summarized below.

Comments

Comment 1, Lines 35-39: The authors state that HspB6 forms oligomers corresponding to dimers and tetramers, but then follow up with a statement suggesting this is somewhat odd because sHsps family chaperones tend to form large oligomers that easily undergo subunit exchange.

Reply

Page 1, lines 38-40, phrase “homo- or heterooligomeric complexes having very dynamic structure and easily exchanging their subunits” has been omitted. The following phrases: “polydisperse assemblies ranging from 10-mer to 40-mer and higher, having very dynamic structure [13-15]. All oligomeric forms possess chaperone-like activity and easily exchange their subunits” have been inserted.

Comment 2, Lines 55-56: Please elaborate on this monomeric mutant. What has been done to promote disrupt oligomerization? Was this reported elsewhere or is it a novel result being reported here?

Reply

A detailed description of the isolation and purification of 14-3-3 monomer is given in Ref. [25], to which we refer in "Materials and Methods"page 2, line 50, the next phrase: "(for example, it forms heterodimers with HspB6)," has been inserted;page 2, lines 53-61, phrases: "them possess chaperone-like activity. It was shown using different target proteins that monomeric form of 14-3-3ζ might have comparable or even higher anti-aggregation activity than either the dimeric form of 14-3-3ζ or HspB6 and HspB5 [24,25]" have been omitted. Instead of it the following phrases: "of isoforms (e and g) are dimeric, whereas other isoforms exist as a mixture of dimer/monomer [23]. It was reported that phosphorylation of Ser58 of certain 14-3-3 isoforms (h, β, and z) in response to activation of different pathways could induce partial dissociation of 14-3-3 dimer (see a review [22]). The structure and functions of the 14-3-3 dimer are well characterized, while much less is known about the properties of the monomeric form of 14-3-3. Earlier it was shown that monomeric 14-3-3ζ has a chaperone-like activity and is stabilized by phosphorylated HspB6 [24]. Previously using different target proteins, we found that monomeric form of 14-3-3ζ might have comparable or even higher chaperone-like activity than either the dimeric form of 14-3-3ζ or HspB6 and HspB5 [25]." have been inserted.

Comment 3, Line 82: Please describe the experimental approach here. Concentrations, temperatures, etc. are provided. However, the thermal aggregation of UV-Phb is unclear. Was the protein previously UV-treated, but then aggregates at 37 degrees Celsius later? If so, is this when HspB6 is added? These details would aid greatly in reader comprehension of this experimental design.

Reply

The test system based on thermal aggregation of UV-irradiated Phb was well studied by us earlier and described in detail in [reference 40: PLoS ONE 2017, 12, e0189125].

But the following text has been inserted in Results, page 2-3, lines 91-96: "For testing anti-aggregation activity of protein chaperones HspB6 and monomeric14-3-3ζ (14-3-3ζm) in the absence and in the presence of 0.1 M Arg UV-irradiated Phb (UV-Phb) was used as a target protein. The studies were carried out using the method of dynamic light scattering (DLS) under physiological conditions: at a temperature of 37 oC and an ionic strength of 0.15 M. The aggregation process was initiated by the addition of UV-Phb to the sample containing Arg, protein chaperones or its mixture." The phrase "A test system based on thermal aggregation of UV-Phb at 37 °C was used at physiological conditions (ionic strength 0.15 M)." has been omitted.

Results, page 3, lines 97, the phrase "obtained using the dynamic light scattering (DLS) method and” has been omitted.

In Materials and Methods, 4.3, page 12, Lines 428-431 two phrases: "The aggregation process was initiated by the addition of UV-Phb to the final volume of 0.45 mL. To study the effects of protein chaperones in the absence or in the presence of 0.1 M Arg on UV-Phb aggregation, the solutions of protein chaperones in buffer were preincubated for 5 min at 37 °C in the cell before the addition of the UV-Phb." have been swapped for a better understanding of the experimental technique.

Comment 4, Line 85: “The initial part of the sigmoidal kinetic curve…” What curve? Please provided figure references if the reader is to follow what is written.

Reply

Results, page 3, line 97 “(Fig. 1A,B; Fig. 3A,B)” has been inserted.

Comment 5, Lines 104,105: “In addition, an increase in the chaperone concentration leads to a decrease in the value of the radii (Rh,2) both in the absence and in the presence of Arg.”

Comparison of Figures 1C and 1D reveal a clear lag phase prior to increase in the hydrodynamic radius. However, conditions lacking Arg seem to reveal slower kinetics. If the increase in Rh,2 as a function of time is due to UV-Phb aggregation, can the authors comment on these apparent kinetic differences? Why does the presence of arginine seem to accelerate aggregate formation?

This observation appears to quantified in Table 1. However, it is still unclear why the aggregate structure would have a lower hydrodynamic radius than the native (or remodeled) state would.

Reply

"In addition, an increase in the chaperone concentration leads to a decrease in the value of the radii (Rh,2) both in the absence and in the presence of Arg." This statement means that the sizes of the hydrodynamic radii Rh,2 begin to increase the later, the higher the concentration of the added protein chaperone. A similar effect is observed both in the presence and in the absence of Arg.

Results, page 3, lines 114-116, the phrase “In addition, an increase in the chaperone concentration leads to a decrease in the value of the radii (Rh,2) both in the absence and in the presence of Arg.” has been replaced by the phrase "In addition, an increase in the concentration of chaperone leads to the fact that the sizes of the hydrodynamic radii of the formed aggregates (Rh,2) begin to increase later both in the absence and in the presence of Arg."

The fact that Arg stimulates the process of UV-Phb aggregation at physiological values of ionic

strength was described in detail in our previous work [reference [36]: Int. J. Biol. Macromol.

2018, 118, 1193–1202.] We suggest that the facilitation of protein aggregation is due to the conformational destabilization of protein by interaction between the guanidinium group of arginine and acidic residues of protein.

Discussion, page 10, lines 327-331, the phrases "and an ionic strength of 0.15 M [36]. It was suggested that in this case the formation of larger aggregates and, as a result, acceleration of UV-Phb aggregation is due to conformational destabilization of UV-Phb molecules in the presence of Arg, which is due to the interaction of the guanidine group of arginine and acidic residues of the protein." have been inserted.

Table 1 presents the Rh* values, that characterizing the size of the nucleus at the end of the nucleation stage. It was shown that for UV-Phb in the absence of additives, the nucleus size is 40 ± 2 nm. In the presence of 0.1 M Arg, protein chaperones or their mixture (Arg + protein chaperone), we observe an increase in the size of the nucleus, which is discussed in detail in the discussion.

Comment 6, Equation 3: Can the authors expand their explanation of this function? Though references are provided, it would make it easier for readers to follow if some explanation were provided. It is quite common to see tryptophan emissions spectra subjected to NLLS analyses, but usually parameters such as λmax or peak area are used, which are model-independent parameters. For this reason, I doubt most readers are going to be familiar with Eq. 3.

Reply

Indeed, Eq. 3 is not well known to most readers. Nevertheless, it seems to us inappropriate to discuss the presented equation in this paper in detail for a number of reasons. This formula was first proposed by Burstein and Emelyanenko in their work [reference 42: Photochem. Photobiol. 1996, 64, 316–320.] In this work it is discussed in detail both the function itself and its applicability for the analysis of tryptophan fluorescence spectra. The use of the presented function for fitting tryptophan fluorescence spectra directly on protein 14-3-3 is described in [reference 43: FEBS Lett. 2011, 585, 1163–1168]. In our work, we simply applied the formula proposed by other specialists for fitting the data we obtained in order to more accurately determine the value of λmax without making any changes to the described methods. Therefore, in this work, only the formula and references to those works where this function is described in detail are given. We believe that this formula can be recommended to all researchers studying tryptophan fluorescence.

Comment 7, Lines 244-245: The authors state that tryptophan residues are located on the surface of the protein in a hydrophilic environment. This statement is an overstatement based on the provided data. The data indicate that the relevant tryptophan residues are in an environment that favors hydrogen bonding to the indole group of tryptophan, but say nothing about solvent accessibility or location. Two equally valid models could be proposed: 1) tryptophan residues undergo an increase in solvent accessibility upon Arg binding or 2) Arg binds adjacent to a tryptophan and directly interacts. However, we have no way of distinguishing between these two models, which requires a more conservative interpretation.

Reply

Results, page 9, lines 260-262. The phrase “tryptophan residues are located on the surface of the protein globule in a hydrophilic environment [45]” has been omitted and the following phrase has been inserted: “the location of tryptophan residues in protein chaperones is favorable for the formation of hydrogen bonds with the indole group of tryptophan.”

Results, page 9, lines 265-266, the phrase “lead to greater accessibility of tryptophan residues to the external environment.” has been omitted and the phrase “perhaps stimulating the formation of new bonds with the solvent or with other amino acid residues of the protein.” has been inserted.

Discussion, page 11, lines 376-377, the phrase “we made the conclusion that the tryptophan residues of HspB6 and 14-3-3ζm are located near the surface of the protein globule.” has been changed on phrase: “we made the conclusion that upon binding of Arg the tryptophan residues of HspB6 and 14-3-3ζm becomes more solvent exposed.”

Discussion, page 12, line 385, the phrase “In addition, the obtained data indicate that the effect of Arg on the conformational state of 14-3-3ζm is more significant compared to HspB6.” has been omitted.

Comment 8, Lines 249-250: Again, this appears to be somewhat overstated. The conclusion is that the effect of Arg on the conformational state of 14-3-3 is more significant than on HspB6. The affinities suggest that binding to HspB6 is stronger. However, the change in fluorescence intensity is greater for 14-3-3 upon Arg binding. This is likely just a consequence of where tryptophan residues are located in overall protein structure. We must remember that the change in overall fluorescence intensity is directly linked to changes in local solvent properties for tryptophan residues. Therefore, we cannot say anything about impact on conformational state beyond the fact that a conformational change does likely occur.

Reply

Results, page 9 , line 266, the phrase “On the basis of the data obtained, it can be concluded that the effect of Arg on the conformational state of 14-3-3ζm is more significant compared to HspB6” has been excluded from the article

Comment 9, Figure 7: Have the authors performed DSC experiments +/- Arg for UV-Phb? It would be interesting to see if Arg also destabilizes this protein, which could make it easier for the chaperones to carry out their respective functions.

Reply

DSC experiments for UV-Phb in the presence or absence of Arg would certainly be useful for describing the changes occurring in the structure of the target protein. Unfortunately, UV-Phb retains only an insignificant part of native structure (7-10%), which was shown earlier (40). Therefore, DSC studies for this protein are not informative.

Comment 10, Line 350: The authors could strengthen their argument for where tryptophan residues are located in overall structure by the generation of homology models that allow for prediction. Alternatively, mapping tryptophan locations onto structures of known homologs could work as well.

Reply

It would be really very interesting to generate models that allow us to predict the location of tryptophan residues in the protein structure, but now we have no opportunity to carry out such work.

Comment 11, Do the authors know what the concentration of Arg is in living cells? Do the KD values reported here approach physiologic levels? This information could be helpful for readers in determining whether the behavior reported here may be relevant to in vivo

Reply

Arg belongs to the category of Amino acids, which are produced in the human body only under certain conditions. With the slightest pathology, the human body does not have time to cope with the production of Arg. At rest, fasting serum L-arginine levels (without additives) are 71 ± 4 nmol/L or 15.1 ± 2.6 mg/mL. Oral administration of an arginine supplement with a dosage of 5 g on an empty stomach promotes an increase in AUC L-arginine (within 5 hours) by 64%. It is believed that in the healthy person should be from 50 to 150 μmoles of Arg. And the values of the Kdis for the 14-3-3ζm and HspB6 are equal to 49 mM, and 10 mM, respectively. But according to the standards developed by sanitary doctors, the daily requirement for Arg consumption is from 4.0 g (19 mM) for children, to 6.0 g (28.5 mM) for an adult.